# Remote Monitoring and Maintenance for Equipment and Production Lines on Industrial Internet: A Literature Review

**Qingzong Li, Yuqian Yang and Pingyu Jiang \***

State Key Laboratory for Manufacturing Systems Engineering, Xi'an Jiaotong University, Xi'an 710054, China
\* Correspondence: pjiang@mail.xjtu.edu.cn

**Abstract:** Monitoring and maintaining equipment and production lines ensure stable production by detecting and resolving abnormalities immediately. In the Industrial Internet, operational technology and advanced information technology are fused to improve the digitalization and intelligence of monitoring and maintenance. This paper provides a comprehensive survey of monitoring and maintenance of equipment and production lines on the Industrial Internet. Firstly, a brief review of its architecture is given, and a reference architecture is summarized accordingly, clarifying the key enabling technologies involved. These key technologies are data collection technologies, edge computing, advanced communication technologies, fog computing, big data, artificial intelligence, and digital twins. For each of the key technologies, we provide a detailed literature review of their state-of-the-art advances. Finally, we discuss the challenges that it currently faces and give some suggestions for future research directions.

**Keywords:** remote monitoring; maintenance; Industrial Internet

## 1. Introduction

### 1.1. Background

Well-maintained equipment and production lines are the basis for regular production in a factory. It is necessary to monitor and maintain the equipment and production lines during the operation process effectively to avoid failures [1–3]. Traditionally, equipment and production lines are manually inspected [4] and maintained after failures arise [5]. However, this strategy cannot avoid the negative impact of equipment downtime on quality and capacity, which entails high costs [6]. With the development of wireless sensor networks [7], advanced communication technologies [8–13], big data [14–16], artificial intelligence [17,18], and digital twins [19], the Industrial Internet emerged, which brings fresh impetus to the monitoring and maintenance of equipment and production lines.

There are several vital issues that should be addressed in the monitoring and maintenance of equipment and production lines on the Industrial Internet. It is a challenge to acquire various types of data from different devices from diverse manufacturers in a factory, as they have different communication protocols that are not compatible with each other. After collecting a vast volume of raw data, it is also a difficult task to store, transfer, and process this data. It is also essential to extract valuable information from these data to determine the health of the equipment and display it to humans to facilitate proper decision-making. Therefore, we present a comprehensive survey of the monitoring and maintenance of equipment and production lines on the Industrial Internet.

### 1.2. Research Methodology for Literature Review

This paper provides a comprehensive investigation of some studies completed on the monitoring and maintenance of equipment and production lines on the Industrial Internet and discusses some research questions and challenges. We started with a brief review of the Industrial Internet's architecture, sorting out the associated key enabling technologies.

Addressing these key technologies, we discuss their latest advances—including data collection technologies, edge computing, communication technologies, fog computing, big data, artificial intelligence, digital twins, data analytics, operation and maintenance (O&M) optimization, and sustainability—enabling researchers to keep up with the pioneers quickly. Then some industrial application examples were demonstrated. We also discussed some of the technical challenges and provided a few suggestions for future research directions.

## 2. Architecture

Architecture is a higher level of abstraction description that helps identify issues and challenges for the monitoring and maintenance of equipment and production lines on the Industrial Internet. There various architectures have been proposed in the recent literature.

Wang et al. [20] proposed a cloud-assisted platform for large-scale continuous condition monitoring based on the Industrial Internet of Things. The platform is a three-tier architecture comprising an edge layer, a platform layer, and an enterprise layer. The edge tier is where data are collected, aggregated, and transmitted. The data are transmitted through the edge gateway to the platform tier for data storage, workflow processing, and other applications. At the enterprise tier, data analysis and mining are applied to support enterprise planning and decision-making. Yang et al. [21] proposed a monitoring platform with a three-layer architecture based on cloud manufacturing, comprising an edge layer, a fog layer, and a cloud layer. At the edge layer, raw data are acquired and preprocessed. The fog layer is devoted to the interconnection of devices and the transmission of data on the one hand and the deployment of trained models on the other. In the cloud, engineers monitor production status, make decisions remotely via screens, and train models for diagnosis and prediction. Li et al. [22] proposed a two-layer Industry 4.0 platform for equipment monitoring and maintaining, consisting of a machine layer and an application layer. The data are collected at the machine layer and then used at the application layer to monitor the equipment conditions, production processes, and product quality. Yang et al. [23] designed an integrated monitoring and maintenance framework for the grinding and polishing robot. The framework is divided into four layers: a physical layer, a key enabling technology layer, a business logic layer, and a data collection and processing layer. The physical layer contains all the devices and sensors. The key enabling technology layer describes the models and algorithms for monitoring and maintenance. The entire business logic is described in the business logic layer. The data collection and processing layer depicts the devices and their corresponding interaction logic for collecting and processing the working condition data in the production line.

From the above literature review, we summarize a reference architecture for monitoring and maintenance of equipment and production lines on the Industrial Internet, as shown in Figure 1. This architecture consists of three layers: the physical layer, the transport layer, and the application layer. The physical layer includes the equipment, sensors, actuators, controller, and data acquisition unit. This layer focuses on data collection and preprocessing, and the key technologies involved in this layer are data collection technology and edge computing. The transport layer contains network transport devices, computing devices, and databases. The main task of the transport layer is data transmission, aggregation, and forwarding, which also involves some data processing. The key enabling technologies in this layer are communication technologies and fog computing. The application layer consists of computing servers, databases, and application servers, whose main functions are data storage, model training, algorithm operation, etc. The key enabling technologies in this layer are artificial intelligence, big data, digital twins, data analytics, O&M optimization, and sustainability. A comprehensive overview of the key enabling technologies in each layer will be presented in the next section.

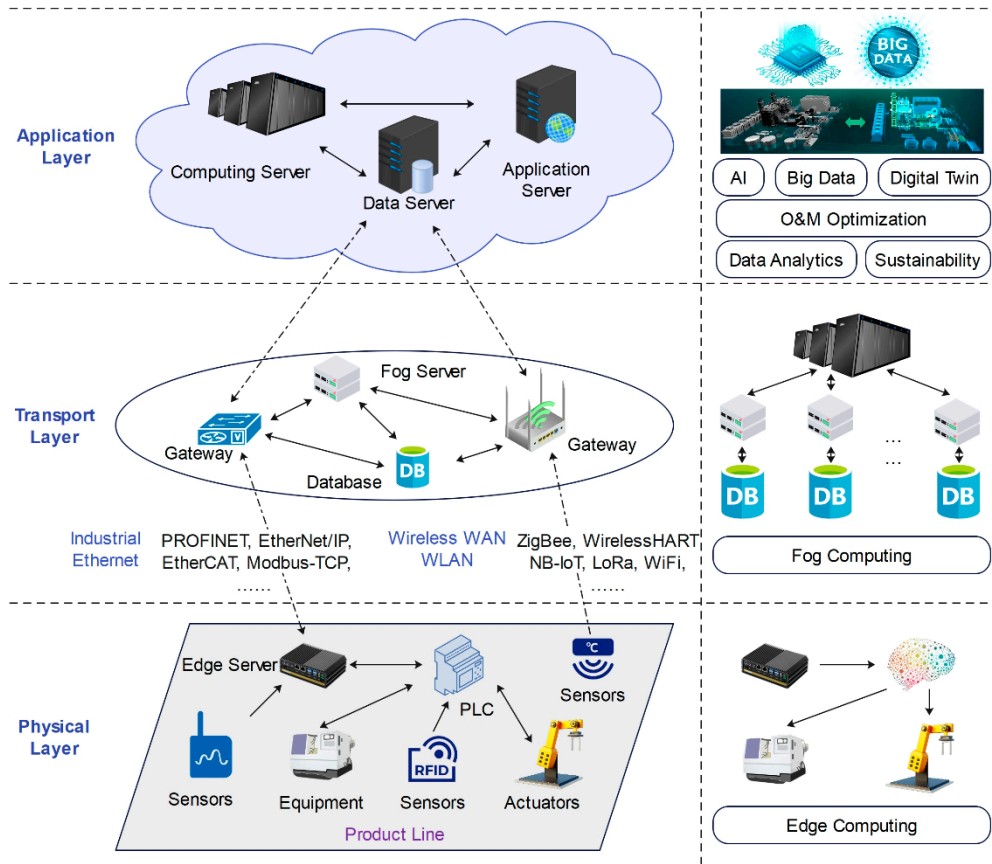

**Figure 1.** Reference architecture.

## 3. Key Enable Technologies

In the age of the Industrial Internet, traditional remote monitoring and maintenance technologies for equipment and production lines are becoming digital and intelligent as they are combined with rapidly evolving information technologies. This section focuses on the key enabling technologies in the above three layers: data collection technologies, edge computing, communication technologies, fog computing, big data, artificial intelligence, digital twins, data analytics, O&M optimization, and sustainability.

### 3.1. Physical Layer

The physical layer includes equipment and data acquisition devices. Its function mainly collects data from the actuators and sensors and processes the acquired data. Therefore, this subsection provides an overview of data acquisition technologies and edge computing.

#### 3.1.1. Data Acquisition Technologies

The data acquisition unit collects data during the production process, including the data collected by sensors and the control data of actuators. From the perspective of physical connectivity, there are two data acquisition modes: wired and wireless.

#### A. Wired Data Acquisition Technologies

The wired data acquisition method is widely used in the industry. Short and Twiddle [24] developed a real-time condition monitoring and fault diagnosis system for large-scale rotating equipment in the water industry. The data acquisition unit of the system contains several temperatures and speed sensors. An ADC converter (AD7856) is used to convert the analog signals from the sensors to digital signals, a C167 microcontroller is used to process the digital signals, and a non-volatile memory chip (EEPROM) is used

for data storage. It communicates with the outside through the RS-232 standard interface. Xia et al. [25] presented an intelligent fault diagnosis system for industrial robot bearings under varying conditions. The vibration from a sensor was collected by a signal acquisition board card (PXIe-4497) and sent to a PXI controller (NI PXIe-8840) together with the joint angle data from the robot controller.

As is shown in Figure 2, wired data acquisition devices usually consist of a data acquisition card, processor, memory, and transceiver. The control data of the equipment can be obtained directly from the controller. Low-frequency sensing data can generally be collected by the controller, but high-frequency sensor data needs to be collected with a dedicated data acquisition card. In general, the data will be processed at the edge, and the processed data will be transmitted to the next layer through the transceiver. The data acquisition card, processor, memory, and transceiver are always integrated into a single device as a data acquisition unit.

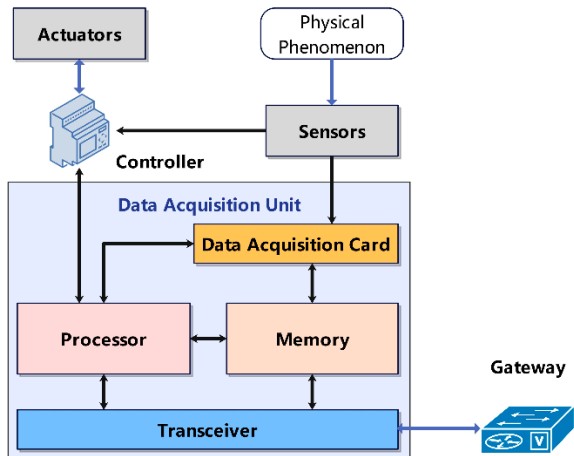

**Figure 2.** Wired data acquisition devices.

B. Wireless Data Acquisition Technologies

At present, wireless data collection methods are developing rapidly [26]. Li et al. [27] proposed a flexible strain sensor based on an aluminum nitride film. Its flexible substrate can stick well to bearings to detect micro-strain, therefore making it suitable for condition monitoring and failure prevention. Sancho et al. [28] proposed a wireless LC sensor to monitor the continuous wear in abradable blades in a paper mill, addressing the issue of wear monitoring in a distributed harsh industrial environment. Ahmed et al. [29] developed an optical camera communication-enabled wireless sensor network to monitor the state of industrial valves. The device consists of an AM2302 sensor to collect the temperature, an ATMEL 1430 TINY85 20SU microcontroller to process the sensor data and modulate the LED, and a transmitter to transfer the data. Walker et al. [30] proposed a method for real-time in-process monitoring of core motion in metal castings. A group of wireless Bluetooth inertial measurement sensors was integrated into the additive manufacturing sand cores to measure the acceleration and rotation during the casting. Lei and Wu [31] designed a wireless device to acquire mechanical vibration signals. The device consists of a high-precision MEMS acceleration sensor, a 16-bit resolution ADC acquisition chip, a high-performance control center (STM32), and a wireless transceiver core (Si4463), which enables high-frequency, high-precision acquisition of vibration signals. Patil et al. [32] proposed an architecture for wireless sensor nodes, arguing that the basic components of a wireless sensor node are a sensor, process unit, memory, transceiver, and battery.

The architecture of the wireless data acquisition devices can be summarized from the above literature, as shown in Figure 3, including sensors, a processor, memory, a wireless transceiver, and a battery. These components are integrated into a single device, commonly referred to as a wireless sensor node.

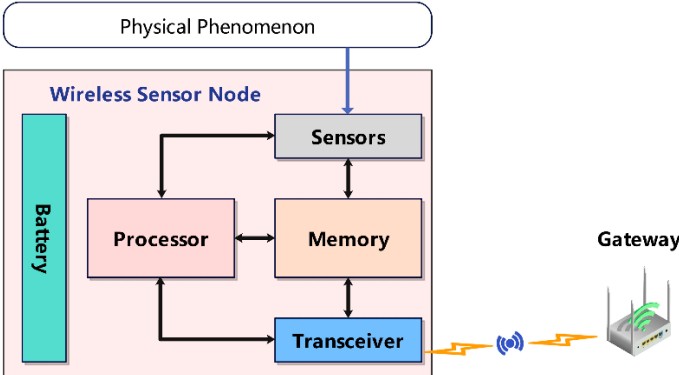

**Figure 3.** Wireless data acquisition devices.

There are numerous advantages of the wireless data acquisition device. It is easy to deploy without any pre-existing infrastructure [33]; the wireless sensor network is capable of covering a pervasive area [34]; the wireless sensor node is convenient to move [7]. Most importantly, wireless sensors can help us collect data in environments where wiring is impossible [35]. Therefore, the application of wireless data acquisition devices in equipment monitoring and maintenance has become a research hotspot.

However, there are several shortcomings in applying wireless sensor data collection methods in equipment and production lines. The wireless sensor node usually requires batteries for power and therefore requires periodic battery replacement, resulting in monitoring interruptions [36]. Even though some self-powered sensors have been developed, they can only be used in specific environments [37]. The real-time performance and efficiency of data transmission remain to be improved [38]. The data are susceptible to interference from the external environment during wireless transmission [39]. These deficiencies limit the application of wireless data acquisition methods in production lines, which require a high degree of reliability and stability of data in real time. It will be the emphasis of future research to address these issues.

### 3.1.2. Edge Computing

A large body of data is collected in production lines. The response time will be too long if all the data are sent to the cloud for processing. Hence, data can be processed at the edge, which reduces response times, increases processing efficiency, and reduces network stress [40–42]. Edge refers to resources and devices near the endpoint along the path between data sources and cloud data centers [43].

Edge computing has already been applied in the remote monitoring and maintenance of equipment and production lines [44]. Zhang et al. [45] developed a cyber-physical machine tool based on edge computing techniques to realize real-time monitoring of the machine tool. The edge devices were deployed on various manufacturing units to process the collected data. The processed data are graphed in real-time with digital twin technology to monitor the process and status of the machine. Edge computing techniques improve the accuracy and capability of virtual machine tools and reduce the mapping latency between physical and digital models. Wen et al. [46] designed a remote monitoring and intelligent maintenance platform for a sewage treatment plant based on edge computing instrumentation. The edge is composed of intelligent instruments and edge servers. Intelligent algorithms and processing units are integrated into intelligent instruments for real-time data collection. The data are sent to an edge server for data quality control, preprocessing, data aggregation, real-time data analysis, decision-making, and data cloud upload. The preprocessed data are used for digital twin modeling and predictive maintenance in the cloud.

The convergence of edge computing and artificial intelligence—called edge intelligence, is becoming a top research priority [18,47]—but there are still many challenges [48].

In the above reviews, the edge was only used for preprocessing of the raw data, and the training and inference of the model were conducted in the cloud. However, with limited bandwidth connectivity between the edge and the cloud, it is almost impossible to achieve real-time local decision-making [49], which is fatal in some hazardous production facilities [50]. Deploying machine learning at the edge is also significantly challenged by the limited computing capacity of edge devices. We discovered that some studies are working on these issues.

Lee et al. [51] presented a predictive maintenance system based on edge computing to maintain and manage motor equipment. The system collects audio data from the motor with an embedded acoustic recognition sensor and then preprocesses the raw data with a preprocessor server. The preprocessed data are transferred to an inference server, which classifies the health of the motor by using a trained model. Both the preprocessing server and the inference server are deployed at the edge. Bowden et al. [52] proposed a hybrid cloud-edge computing-based framework called SERENA for predictive maintenance. A machine-learning model was built, trained in the cloud, and then pushed to the edge device. In the edge device, the statistical characteristics of the raw data from the sensors are calculated first, and then machine learning models are used to diagnose the failures of machines.

From the above studies, a general edge intelligence architecture can be summarized, as shown in Figure 4. The raw data are preprocessed at the edge, and the results are sent to the cloud. In the cloud, the preprocessed data can be applied directly to remote monitoring and maintenance of equipment, as it can also be used to train machine learning models. The trained model can be deployed at the edge for inference of anomalies and consequently signaling an alarm.

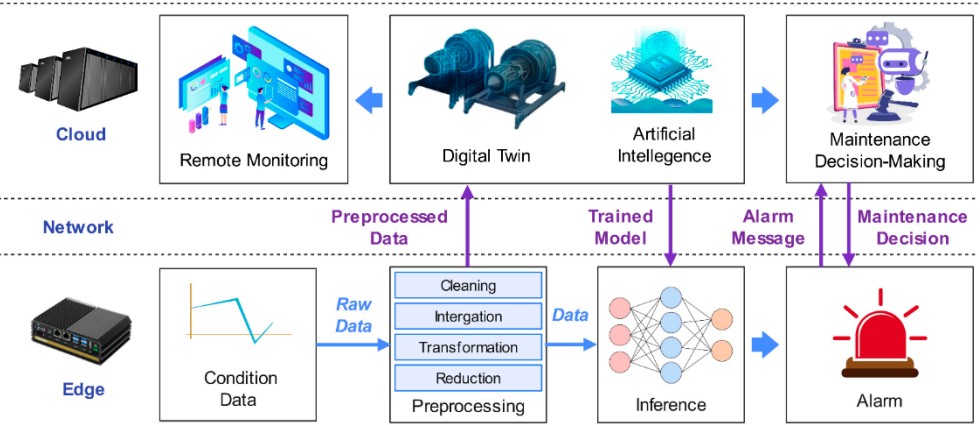

**Figure 4.** General edge intelligence architecture.

### 3.2. Transport Layer

The transport layer is mainly responsible for data transmission, aggregation, and forwarding. Hence, a review of communication technologies is necessary. As the volume of data increases, there is an additional need to store and compute data at this layer. Thus, an overview of fog computing is provided in the following paragraphs.

### 3.2.1. Communication Technologies

There are also two modes of data transmission: wired and wireless. This section reviews both wired and wireless communication protocols. In addition, because the Open Platform Communication Unified Architecture (OPC UA) enables the interconnection of devices under different protocols, it is also reviewed in this section

A. Wired Communication Technologies

Wired communication is the most universal and widely used mode. Wired communication protocols comprise Fieldbus and industrial Ethernet. Figure 5 illustrates the classification of the wired communication protocols.

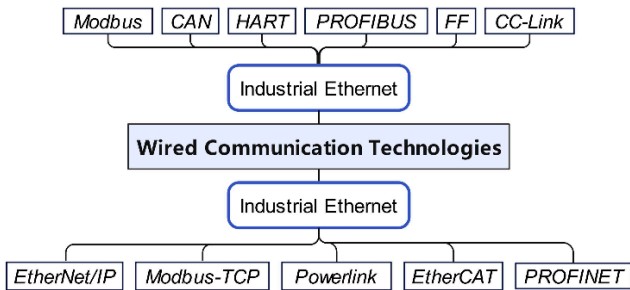

**Figure 5.** Wired communication technologies.

Fieldbus technology has been applied in industrial automation since the 1970s with a long history [53]. Because different companies developed their own products, numerous standards were created. The most frequently implemented Fieldbus protocols are Modbus [54], Controller Area Network (CAN) [55], Highway Addressable Remote Transducer (HART) [56], INTERBUS [57], PROFIBUS [58], Foundation Fieldbus (FF) [59], Control & Communication Link (CC-Link) [60], etc. Fieldbus technology is being phased out in the Industrial Internet age because devices with different protocols cannot communicate with each other.

With the development of Ethernet technology, Industrial Ethernet is gradually replacing the Fieldbus as the solution for the interconnection of equipment [61]. Protocols for Industrial Ethernet include EtherNet/IP [62], Modbus-TCP [63], Powerlink [64], EtherCAT [65], PROFINET [66], et al. Compared to Fieldbus technology, Industrial Ethernet has the advantages of fast transmission rates, long transmission distances, better interoperability, flexible topology, and easy integration [67]. However, the Industrial Ethernet does not solve the problem of interconnection between devices of different protocols either.

B. Wireless Communication Technologies

Although wired communication has the advantage of high reliability and low latency in the industry, there is still a place for wireless communication methods. Wireless communication methods are often used in environments where wiring is extremely hard, such as hazardous areas, moving equipment, etc. [68]. Figure 6 demonstrates the popular wireless communication protocols.

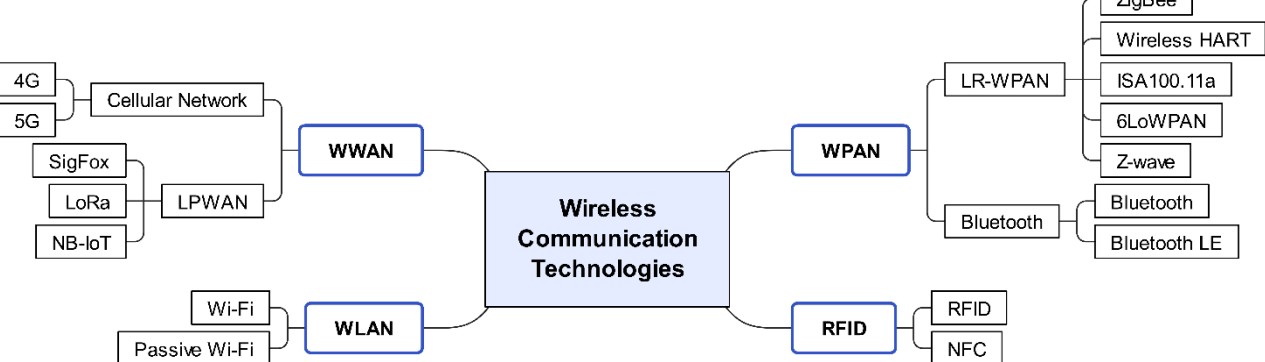

**Figure 6.** Wireless communication technologies.

A wireless wide area network (WWAN) is a telecommunications network that extends over a large geographic area. One approach to implementing WWAN is mobile telecommunication cellular network technologies, sometimes called Mobile Broadband. Technologies that can be used in the Industrial Internet include 4G [69] and 5G [70]. One of the drawbacks of the cellular network is its high power consumption. Hence the low-power wide area network (LPWAN) was proposed, which is a promising solution for remote and low-power Internet of Things [71]. The major LPWAN technology solutions include SigFox [13], LoRa [72], and Narrowband IoT (NB-IoT) [9]. A wireless local area network (WLAN) is a telecommunications network that links two or more devices within a limited area, such as a workshop or a production line [73]. The most widely used WLAN in the Industrial Internet is often known as Wi-Fi, based on the IEEE 802.11 standard [74]. To solve the problem of high power consumption of Wi-Fi, a low-power Wi-Fi called 'passive Wi-Fi' has been proposed recently [75]. A wireless personal area network (WPAN) is a telecommunications network within an individual's workspace [76]. The most widely used WPAN technology is Bluetooth [77] and Bluetooth Low Energy (Bluetooth LE) [78], based on the IEEE 802.15.1 standard, which has been used in the industry for a long time. However, the energy consumption of Bluetooth technology is relatively high, so the low-power, low-cost has been proposed, called low-rate wireless personal area networks (LR-WPAN), is also commonly used in industry [79]. The common LR-WPAN technologies are ZigBee [80], WirelesHART [81], ISA100.11a [82], 6LoWPAN [83], and Z-Wave [84]. Radio-frequency identification (RFID) [85] is a technology that automatically reads the information on a tag with electromagnetic fields, and Near Field Communication (NFC) [86] technology is developed based on it.

A qualitative and quantitative comparison between the wireless communication technologies is given in Table 1. In terms of the requirements of industrial applications, we have compared the performance of various wireless communication technologies from three perspectives: coverage range, power, and data rate. Cellular networks are suitable for the transmission of large amounts of data over long distances but require a stable energy supply. LPWAN is ideal for transmitting small amounts of data over long distances and benefits from low energy consumption. Wi-Fi is appropriate for transferring large amounts of data over short distances but consumes more energy. While passive Wi-Fi reduces energy consumption, it also reduces data transfer rates. Bluetooth and LP-WPAN are used for low-rate data transmission over short distances. Bluetooth Classic is slightly faster but consumes more energy; Bluetooth LE has lower power consumption but also lower speed. RFID and NFC require reading information from a tag at a short range, so they are commonly used for identifying and tracking objects. Every wireless communication technology has its own characteristics, and it is necessary to choose the right technology for application according to real industrial scenarios [87].

C. Open Platform Communication Unified Architecture

As can be seen from the previous review, there are different communication protocols in industrial applications, and devices with different protocols cannot interconnect with each other. However, in real production scenarios, a company's equipment always comes from various manufacturers and supports different communication protocols. The interconnection and interoperability of different devices must be achieved in the Industrial Internet [88]. The advent of Open Platform Communication Unified Architecture (OPC UA) provides a solution to this problem [89].

OPC UA is a cross-platform, open-source IEC 62,541 standard developed by the OPC Foundation that is used for the reliable, secure, and interoperable transfer of data [90]. The IEC 62,541 standard consists of the following parts: part 1—Overview and Concepts; part 2—Security Model; part 3—Address Space Model; part 4—Services; part 5—Information Model; part 6—Mappings; part 7—Profiles; part 8—Data Access; part 9—Alarms and Conditions; part 10—Programs; part 11—Historical Access; part 12—Discovery and Global Services; part 13—Aggregates; part 14—PubSub. Parts 1 to 7 specify the core functions of

OPC UA that define the modeling approach in the address space and the services associated with it. Parts 8 to 13 are the access type specifications of OPC UA. Part 14 enables OPC UA to support the Publish/Subscribe communication mode, improving the scalability of the system. There has been a lot of research on OPC UA.

**Table 1.** Comparison between wireless communication technologies.

| | Technology | | Cover Range | Data Rate | Power |
|---|---|---|---|---|---|
| WWAN | Cellular Network | 4G | **Long-Range** 10 km | **High** 100 Mbps | **High** |
| | | 5G | **Long-Range** 1 km | **High** 10 Gbps | **High** |
| | LPWAN | SigFox | **Long-Range** 10 km (urban), 40 km (rural) | **Low** 100 bps | **Low** |
| | | LoRa | **Long-Range** 5 km (urban), 20 km (rural) | **Low** 50 kbps | **Low** |
| | | NB-IoT | **Long-Range** 1 km (urban), 10 km (rural) | **Low** 200 kpbs | **Low** |
| WLAN | Wi-Fi | | **Short-Range** 50 m | **High** 1 Gbps | **High** |
| | Passive Wi-Fi | | **Short-Range** 30 m | **Low** 11 Mbps | **Low** |
| WPAN | Bluetooth | Bluetooth Classic | **Short-Range** 100 m | **Low** 3 Mbps | **Moderate** |
| | | Bluetooth LE | **Short-Range** 100 m | **Low** 2 Mbps | **Low** |
| | LR-WPAN | ZigBee | **Short-Range** 100 m | **Low** 250 kbps | **Low** |
| | | WirelessHART | **Short-Range** 200 m | **Low** 250 kbps | **Low** |
| | | ISA100.11a | **Short-Range** 600 m | **Low** 250 kbps | **Low** |
| | | 6LoWPAN | **Short-Range** 100 m | **Low** 250 kbps | **Low** |
| | | Z-wave | **Short-Range** 100 m | **Low** 100 kbps | **Low** |
| RFID | RFID | | **Short-Range** 100 m | **Low** 400 kbps | **Moderate** |
| | NFC | | **Short-Range** 0.04 m | **Low** 400 kbps | **Low** |

Liu et al. [91] proposed a cyber-physical machine tools (CPMT) platform based on OPC UA and MTConnect. The authors developed an MTConnect to OPC UA interface to solve the interoperability problem between OPC UA and MTConnect, which converts the MTConnect information model and data into OPC UA counterparts. This platform has enabled standardized, interoperable, and efficient data communication between machine tools and various software applications. Kim and Sung [92] designed an OPC UA wrapper that allows UA clients to access legacy servers with OPC Classic interfaces seamlessly. The OPC UA wrapper consists of a UA server and a classic client that interact with each other through shared memory and semaphore. Martinov et al. [93] presented an OPC UA system to monitor the working process of CAN servo drives. The system consists of servo drives, a motion controller, a CNC kernel, and an OPC UA Server. Servo drives are connected to the motion controller via the CAN bus. The CNC kernel collects the whole data from the motion controller and sends it to the OPC UA server. The OPC UA server will convert this data into a uniform format via the information model so that the OPC UA client can

monitor the servo drive remotely. Wang et al. [94] proposed a tool condition monitoring methodology considering tool life prediction based on the Industrial Internet. The OPC UA server collects data from the machine via a private protocol and then converts the data via the information model internally. The converted data are used for tool condition monitoring and life prediction.

A generic OPC UA-based device data acquisition framework can be summarized from the literature review above, as shown in Figure 7. With the implementation of the OPC UA server, sensors under different protocols can communicate with cloud applications. First, the data from the sensor is transferred to the OPC UA server. The information model of the corresponding sensor is created in the OPC UA server. Data are encoded into the standardized message through the information model. Finally, the cloud applications can process these standardized messages in the OPC UA client to enable remote monitoring and maintenance of the equipment.

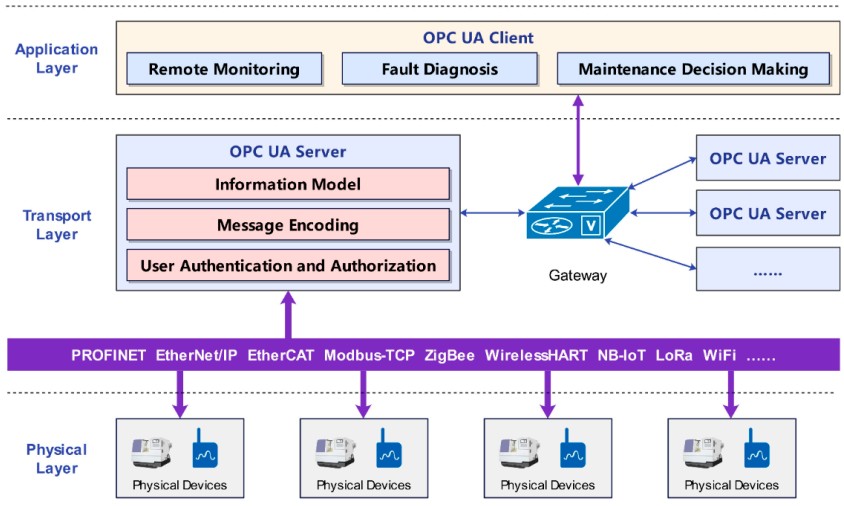

**Figure 7.** OPC UA-based data acquisition framework.

### 3.2.2. Fog Computing

Because of the insufficient computing capability of the devices at the edge, it is impossible to perform complex calculations on the data at the edge while guaranteeing the real-time performance of the transmission [95]. However, with the application of artificial intelligence technology in the field of equipment maintenance becoming more advanced, higher demands are placed on the computing capability of the system [96]. Transmitting all the raw data to the cloud for calculation again suffers from the problem of excessive data volume [97]. Therefore, processing the data at the transport layer becomes a solution for these issues.

Fog computing was considered an implementation of edge computing in previous studies [98]. However, fog computing has evolved into a new computing paradigm. It incorporates the concept of edge computing and provides a structured middle layer between the edge and the cloud, bridging the gap between the Internet of Things and cloud computing [99]. The fog node is not necessarily directly connected to the end device; it can be located anywhere between the end device and the cloud [100]. In the Industrial Internet, fog computing and federated learning are usually used in a fusion, as fog computing provides greater computational capacity for artificial intelligence applications.

Liu et al. [101] proposed a wireless signal classification framework based on federated learning. The raw data are processed by frequency reduction and sampling pretreatment, and its intelligent representation is obtained. The intelligent representation is put into the neural network in the node for training. The loss function of each node is sent to the aggregator and aggregated. Then the result is fed back to each node and used for gradient optimization to achieve global aggregation. It can solve the problem of reduced

signal classification rates caused by different types of mixed signals in complex industrial environments and protect the privacy of industrial information. Brik et al. [102] proposed a disruption monitoring system based on fog computing and federated learning to monitor production for interruptions. The system collects position and movement data from manufacturing resources (workers, robots, equipment, etc.) via cameras and then transmits the data to the fog nodes for calculation. Each fog node trains a local prediction model and then transfers the model weights to the cloud server only. Federated averaging (FedAvg) algorithm is used in the cloud server to aggregate all local models to generate a global model. Global models are deployed to the fog nodes to predict the location of manufacturing resources. Production is considered to be interrupted once the measured position is found to be inconsistent with the predicted position. This failure information will be transmitted to the cloud, where the production tasks will be rescheduled via a rescheduling algorithm. The tasks after rescheduling are then transmitted to the manufacturing resources for production adjustment.

A fog computing framework fusing federated learning can be summarized from the above literature, as illustrated in Figure 8. The raw data are collected from the devices at the edge, and the collected data can be preprocessed or transmitted directly to the fog nodes. In the fog node, the initial global model is first downloaded and then trained locally. The local model in each fog node will be transferred to the cloud after training. In the cloud, the local models from each fog node will be aggregated to generate a new global model, which can be deployed to the fog node for equipment maintenance.

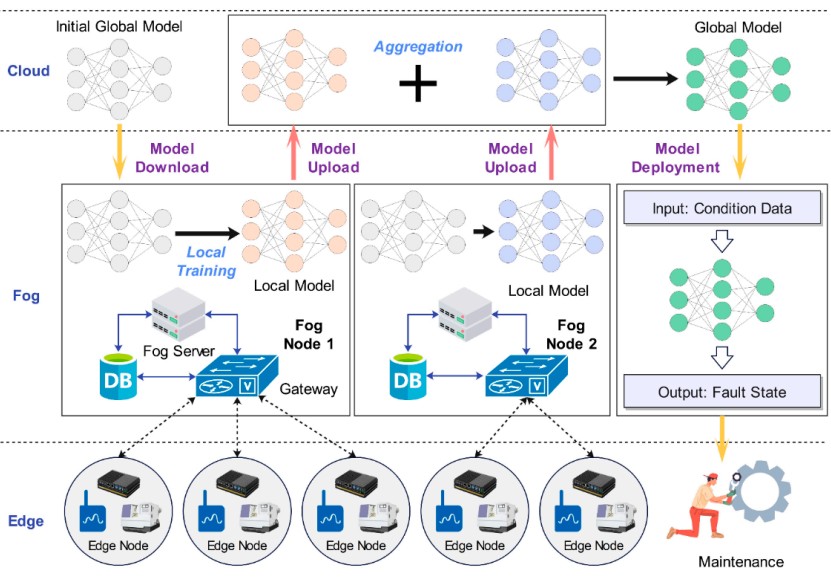

**Figure 8.** Fog computing framework fusing federated learning.

*3.3. Application Layer*

The application layer runs in the cloud with powerful computing capabilities for the final processing of the transmitted data. The results of the calculations can be presented directly to the engineers to make the correct decisions. The following is an overview of six perspectives: big data, artificial intelligence, digital twins, data analytics, O&M optimization, and sustainability.

3.3.1. Big Data

In the Industrial Internet, massive amounts of data are generated by a large number of sensors and controllers [103]. These data are characterized by their large volume, fast transmission speed, and variety of types and are known as 'big data' [14,15]. Traditional approaches to data storage and computation are unable to process such large volumes of data. Hence a new computing paradigm is required.

Yu et al. [104] presented a global manufacturing Big Data ecosystem for predictive maintenance, which involves the acquisition, storage, processing, and visualization of data. It is applied to detect abnormal patterns in the syngas reciprocating compressor. The system continuously collects signals such as vibration, temperature, pressure, and speed from the turbine syngas compressor through hundreds of sensors, averaging approximately 57 million entries per day. After such a large amount of data has been collected, it is replicated in triplicate and stored randomly on cloud nodes via an optimized Hadoop Distributed File System (HDFS) to avoid data loss. During data analysis, the data are first converted into DataFrame format and stored in the Apache Hive Central Data Warehouse and MapR Binary Database, respectively. The data are then computed using the MapReduce-based distributed PCA algorithm with Apache Spark as the data processing engine to enable the identification of equipment failures. The identification results are presented to the engineers on a visualization screen. Wan et al. [105] proposed a Spark-based parallel ant colony optimization (ACO)-K-means clustering algorithm for fault diagnosis of large amounts of rolling bearing operating condition monitoring data. The collected 119.8 GB of raw bearing vibration data was stored in the HDFS. The data are clustered on the Spark efficient computing platform with the ACO-K-means clustering algorithm to obtain a fault diagnosis model. The results demonstrate that the big data computing framework can improve the efficiency of model computation and fault diagnosis.

In the acquisition of big data, the high concurrency problem must be solved because a large amount of data arrives at the database simultaneously, causing blocking. Park and Chi [106] introduced a high throughput data ingestion system for machine logs in the manufacturing industry. Machine log stream data from a group of milling machines are first sent to a set of pre-assigned distributed buffers called Topic. Apache Kafka manages these Topics so they can be stored in the database orderly. Sahal et al. [107] studied a big-data-based predictive maintenance case in wind energy. Wind farms are geographically distributed, and data from wind turbines need to be aggregated with a standard data storage model. Therefore, RabbitMQ is well suited to solve this problem with its federated queues. RabbitMQ is a distributed queuing management technology based on the Advanced Message Queuing Protocol (AMQP), which can ensure receiving data from sensors in the correct order. Under the AMQP protocol, the publisher's messages are transferred to the exchange. The exchange distributes the received messages to the bound queues according to the routing rules. Finally, the AMQP agent delivers the messages to the consumers who have subscribed to this queue. Consumers can also retrieve these messages by themselves as needed. Liu et al. [108] employed the publish/subscribe communication protocol Message Queuing Telemetry Transport (MQTT) to realize the data exchange between different equipment. After the publisher's message is transmitted to the MQTT Broker, it is routed directly to the subscriber and is not stored in the queue. Therefore, MQTT has low energy consumption and is perfect for small devices.

The collected industrial big data are massive, multi-source, and heterogeneous—containing structured, semi-structured, and unstructured data—thus creating a huge challenge for data storage. Traditional relational databases are excellent for storing structured data. However, with the explosive growth of data volume, Structured Query Language (SQL)-based information query has become unable to meet the demand due to its inherent limitations in terms of scalability and fault tolerance [109]. Hence, NoSQL databases are gradually becoming the solution for storing big data. Martino et al. [110] compared the performance of three popular NoSQL Database Management Systems—namely Cassandra, MongoDB, and InfluxDB—in storing Industrial big data. The results show that InfluxDB has the best performance because the data streams from industry devices can be considered a collection of time series. In order to support Online Analytical Processing (OLAP) of big data rather than just storing data, data warehouse has been proposed. Silva et al. [111] demonstrated a logistics big data warehouse for the automotive industry. The data warehouse stores current and historical logistics data to support real-time monitoring of logistics status and online prediction of on-time delivery using machine

learning algorithms. The data warehouse dramatically improves the efficiency of online data analysis. Data lake is proposed to store heterogeneous data from different sources. Munirathinam et al. [112] designed a semiconductor manufacturing data lake with Hadoop. It stores all data from multiple business units, including batch data, process data, and quality data from wafer production. The authors demonstrate a visual analysis of the data with hive tables and tableau. Various organizations of the company—such as the Manufacturing and Quality departments—use it to benchmark across different design IDs and fabs, and set yield/quality ramp and maturity targets for newer product generation. A data lake is a repository of data from disparate sources stored in their original format and can contain everything from relational data to JSON documents to PDFs to audio files. The data can be stored without conversion leading to highly efficient. Its flexible character allows business analysts and data scientists to look for unexpected patterns and insights. Data lakes are becoming the most advanced solution for big data storage.

The above literature shows that big data computing frameworks mainly address the problem of distributed storage and computation of massive amounts of data, as shown in Figure 9. The enormous amount of data collected from the equipment and production lines will be stored. The dominant distributed storage system is HDFS. The data are then processed on the computing cluster with Hadoop MapReduce, Spark, and other computing engines. They employed machine learning, data analysis, and other approaches to monitor equipment and production lines remotely.

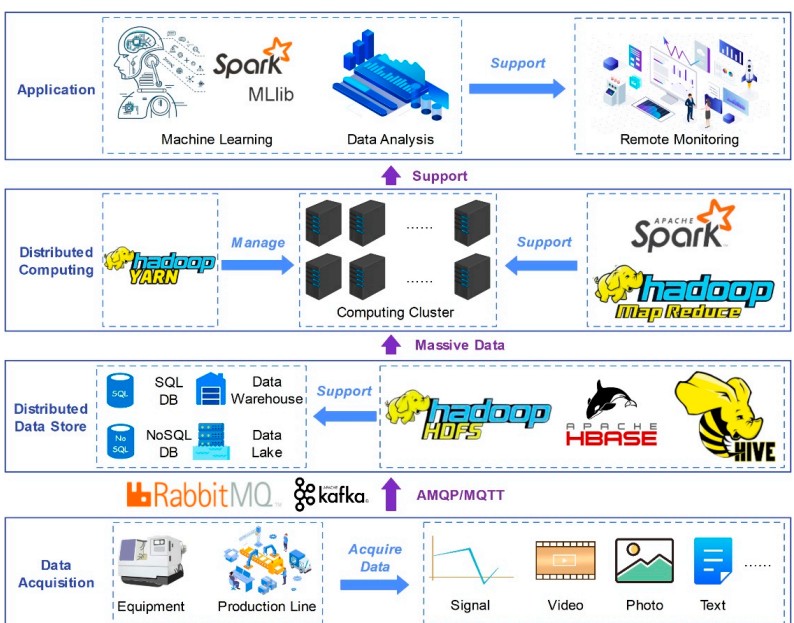

**Figure 9.** Big data application framework.

### 3.3.2. Artificial Intelligence

Fault diagnosis and anomaly detection are the main applications of artificial intelligence in equipment maintenance. Fault diagnosis can identify the reason for a fault to occur. Anomaly detection can only determine the occurrence of a fault, but it has advantages in terms of dataset generation. This subsection takes the literature review from these two perspectives.

### A. Fault Diagnosis

Pursuing the relationship between monitoring data and machine health states is always a widespread concern in machine health management, and fault diagnosis plays a significant role in solving such issues [113]. Machine learning-based fault diagnosis

has been widely used in the monitoring and maintenance of equipment and production lines [114].

Han and Li [115] developed a novel out-of-distribution (OOD) detection-assisted trustworthy machinery fault diagnosis method. At first, a deep ensembled fault diagnosis system is established by integrating multiple deep neural networks. Then, a trustworthiness analysis is performed with an uncertainty-aware depth ensemble to detect OOD samples and give warnings about potentially unreliable diagnoses. Lastly, the deep ensembles' prediction and uncertainty are carefully considered to achieve trustworthy decisions. The proposed method was validated with a wind turbine fault diagnosis case and a gearbox fault diagnosis case. The wind turbine fault dataset consists of one set of normal data and three sets of faults, each with 1000 samples. The gearbox fault dataset consists of one set of normal data and four sets of fault data, each with 1000 samples. The result demonstrates that it exhibits significant advantages in diagnosing OOD samples and obtaining trustworthy fault diagnosis results. Liu et al. [116] proposed a deep feature-enhanced generative adversarial network (GAN) for rolling bearing fault diagnosis. A new generator objective function integrated with a pull-away function was designed to avoid mode collapse phenomena and improve the stability of the GAN. A self-attention mechanism is used in the GNN to enhance the learning of the features of the original vibration signal. In order to guarantee the accuracy and diversity of the generated samples, an automatic data filter was constructed. At last, a convolutional neural network is added as a classifier for fault diagnosis. The method was validated on a rolling bearing vibration signal dataset from an electric locomotive. The dataset contains one set of normal data with a 126,000 sample size and five sets of fault data with a 12,600 sample size. The results demonstrate the better performance of this method in unbalanced sample fault diagnosis. Ferracuti et al. [117] proposed a fault diagnosis algorithm for rotating machinery based on the Wasserstein distance. The authors extracted frequency- and time-based features from the vibration signals and then considered the Wasserstein distance in the learning phase to differentiate the different equipment operating conditions. The statistical distance-based fault diagnosis technique permits obtaining an estimation of fault signature without training a classifier. Therefore, it is very efficient and can be used for embedded hardware. This algorithm can solve the problem of fault diagnosis for rotating machinery at low signal-to-noise ratios and different operating conditions. It can also be applied to system monitoring and prognostics, allowing for predictive maintenance of rotating machinery. Ferracuti et al. [118] studied the problem of defect detection and diagnosis of induction motors based on motor current signature analysis. The researchers estimate the probability density functions of data related to healthy and faulty motors with a Clarke–Concordia transformation and kernel density estimation. Kullback–Leibler divergence is used as an index for the automatic identification of defects because it identifies the dissimilarity between two probability distributions. Fast Gaussian transform improves kernel density estimation. This method has a low computational cost and enables real-time quality control at the end of the production line. The experiments show that the proposed method can detect and diagnose different induction motor faults and defects.

As can be seen from the above reviews, fault diagnosis methods require the training of machine learning models on labeled fault datasets. The trained model can identify the occurrence of a fault and diagnose the type of fault according to the acquired data. However, it is difficult to collect sufficient fault data in a real production scenario, so there are limitations in the application of this method.

B. Anomaly Detection

Equipment failures are rare in real production scenarios, so we can only obtain a tiny amount of equipment failure data. Therefore, it is impossible to acquire enough fault data to train the machine learning models, which brings a huge challenge for the application of machine learning techniques in predictive maintenance [119]. Anomaly detection algorithms are an effective way to solve this problem [120].

Zhao et al. [121] proposed a one-class classification model based on extreme learning machine boundary (ELM-B) to detect bearing failures. The dataset is a NASA-bearing dataset provided by the Center for Intelligent Maintenance Systems (IMS) at the University of Cincinnati. The model is a single-layer feed-forward neural, with an input layer, a hidden layer and an output layer. The researchers calculated the RMS, kurtosis, peak–peak, crest factor, and skewness of the healthy bearing vibration signals in the dataset as inputs to the model. The model is trained to produce 1 at the output. The vibration signals of the bearings are collected by sensors and fed into the trained model. If the output is not equal to 1, it is assumed that a fault has occurred. Tanuska et al. [122] proposed an anomaly detection algorithm for detecting anomalies in conveyor carrier wheel bearings in automotive assembly lines. The researchers collected 16,000 bearing temperature data, of which there were only 18 abnormalities. They designed a multi-layer perceptron (MLP) with 13 neurons in the input layer, 18 neurons in the hidden layer, and 2 neurons in the output layer. The MLP can detect bearing anomalies based on the minimum and average temperature of the bearing. Kähler et al. [123] presented an anomaly detection approach based on a convolutional autoencoder (CAE) to detect surface defects in aircraft landing gear components. The CAE consists of an encoder and a decoder. The encoder comprises an input layer and a convolutional layer for compressing the image. The decoder reconstructs the compressed representation of the input using transposed convolution and convolution layers. The researchers collected 600 defect-free images and 300 defective images of aircraft landing gear surfaces. From this sample, 500 of the defect-free images are randomly selected for training, while the remaining 100 defect-free images and 300 defective images are used for testing.

From the above literature, it can be concluded that anomaly detection approaches can contribute to solving the problem of insufficient fault samples in the industry because the method only requires normal data for the training. However, the shortcoming of anomaly detection is that the exact cause of the fault cannot be identified.

### 3.3.3. Digital Twin

After the collected data has been processed, the results should be presented to the engineers for monitoring. Traditional monitoring methods include simple charts, pictures, two-dimensional electronic kanban or videos, etc. These methods suffer from poor visibility, low interactivity, and limited scalability, making it difficult for engineers to have a comprehensive understanding of the operating conditions of equipment and production lines. Digital twin technology can solve these problems to a certain degree [124,125].

Fan et al. [126] proposed a generic architecture and implementation method for digital-twin visualization. Data collected in the production line is encapsulated in Automation Markup Language (AML), a digital twin data exchange format, and then cached, managed, and transformed in real-time by Cyber Engine for visualization. Digital mock-up (DMU) enables the physical data of a production line to be constructed as virtual 3D scenario motions and state changes. The digital twin can present complete and visual information about the production line to the engineers, improving their decision-making. Fera et al. [127] proposed a novel digital twin framework to evaluate the production line performance. This framework collects data on the body posture and working hours of workers in production with a wearable sensor. The digital twin of the worker is created by binding human motion data to the digital human model via a special interface. This human motion data will be tied to a digital model of a human to generate a digital twin model of the worker. The management team can evaluate the efficiency of the production line accordingly and optimize the assignment of production tasks. Liu et al. [108] proposed a digital twin-based cyber-physical production system (CPPS). Digital geometry models of 3D printing lines are encapsulated using web technology. An ontology-based information model was designed to bind the data to the geometry model for 3D visualization and remote control of the production line.

As shown in Figure 10, the digital twin is combined with data collection techniques, artificial intelligence, and big data computing to map equipment and production lines in the physical space to the digital space. It allows remote monitoring of equipment and production lines, giving a comprehensive insight into the production scenario so that the correct decisions can be made.

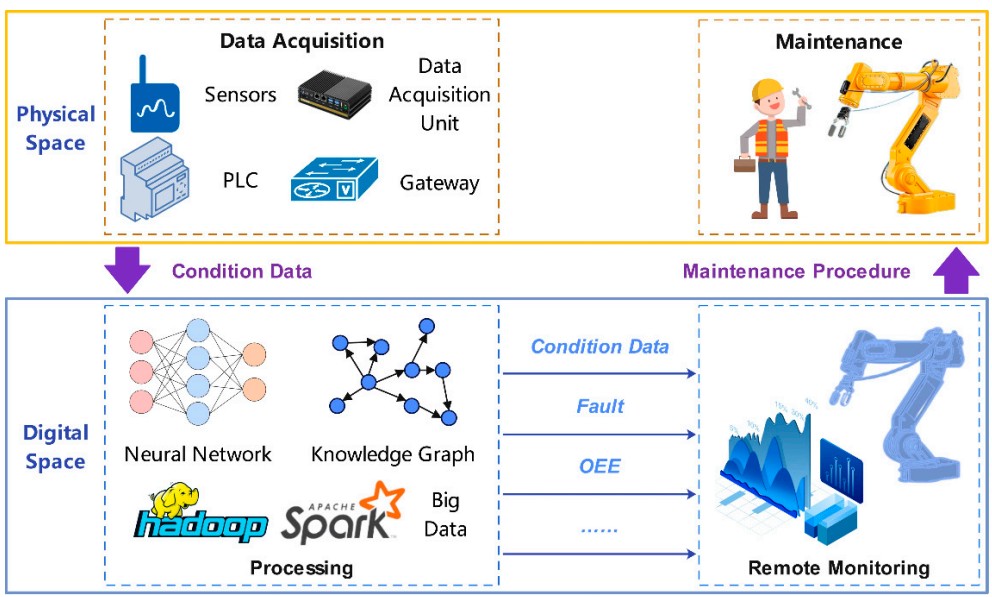

**Figure 10.** Digital twin.

### 3.3.4. Data Analytics

Data analytics is the process of collecting, managing, processing, analyzing, and visualizing evolving data [128] which has a wide range of applications in the Industrial Internet.

Zuo et al. [129] proposed an Internet-of-Things (IoT) and cloud-based novel approach for product energy consumption evaluation and analysis (ECEA). Data related to product energy consumption is dynamically collected in real-time. The system analyzes the energy consumption of the products in the pre-production, production, and post-production stages—including transportation energy consumption, processing energy consumption, auxiliary energy consumption, usage energy consumption, etc. A design solution for a bearing bracket in a toy aircraft was optimized with this method and gave a design solution with minimum energy consumption, qualified functional quality, and within-budget cost. Zhong et al. [130,131] proposed a real-time big data analytics framework to monitor intelligent manufacturing shop floors. Researchers tracked workpieces in production in real-time with RFID. RFID-cuboids were introduced to represent logistics information, thus mining trajectory knowledge and associated indexes for evaluating various manufacturing objects such as workers and machines. The steps of knowledge mining include data cleaning, compression, classification, and pattern recognition. This knowledge supports differentiated decision-making, for example, logistics planning, production planning and scheduling, as well as enterprise-oriented strategy.

The above literature shows that the collected data needs to be cleaned and compressed. Data mining, statistical analysis, and other methods are applied to obtain useful knowledge for production monitoring and optimization.

### 3.3.5. Operations and Maintenance Optimization

Operations and maintenance optimization is a significant application in industrial production. There are several studies that have been conducted to address this issue.

Yang et al. [132] developed a weather-centered opportunistic O&M framework to enable a flexible maintenance resource allocation for wind turbines. This framework quan-

tifies the negative (delays caused by severe wind conditions) and positive (maintenance opportunities) impacts of wind conditions on production and maintenance processes. The advantage of this framework is the consideration of providing additional maintenance opportunities when wind velocities are too small to keep the turbines running. Maintenance downtime and production losses are significantly reduced because these spare times are fully utilized. The authors developed a renewal scenario for turbine components, built a maintenance cost model, and derived the optimal maintenance age for minimizing the maintenance cost of wind turbines with sufficient maintenance resources. An improved performance-based contracting (PBC) model was established to capture the comprehensive effect of both production and maintenance processes. This framework collects wind turbine condition data and wind velocity data for real-time monitoring so that maintenance tasks are optimized with the support of the PBC. A case study shows that this framework is more flexible in resource allocation, significantly reducing maintenance costs and increasing revenue. Hu et al. [133] proposed a joint decision-making strategy for job scheduling and preventive maintenance (PM) planning for a two-machine flow shop with resumable jobs, where both job-dependent operating conditions (OC) and imperfect maintenance (IM) are considered. A hybrid processing time model is built to obtain the optimal sequence when the failure rate of a machine is constant under a fixed OC. The authors presented a joint optimization model for job scheduling and PM planning when the machine failure rate is time-varying at a fixed OC and calculated it with a genetic algorithm. The advantage of this method is that it considers the OC data collected during the production. The results demonstrate that the approach is effective in reducing production completion time and also in reducing the frequency of failures.

From the above review, it can be concluded that the current research for O&M optimization not only considers the maintenance tasks and production tasks of the equipment but also the operating conditions of the equipment. Real-time monitoring of working conditions enables further optimization of production and maintenance task scheduling issues, improving productivity and reducing maintenance costs.

### 3.3.6. Sustainability

Sustainability is an essential issue in the context of climate change, where industrial production plays an important role. There are several applications for improving sustainability in manufacturing with remote monitoring and maintenance on Industrial Internet.

Rojek et al. [134] proposed a digital twins system for manufacturing and maintenance sustainability. The authors obtained real data from some companies engaged in eco-design, process planning, and process supervision. Several artificial intelligence models were built based on this data and fused into the digital twins. The digital twins monitor production processes with artificial intelligence models for process parameter optimization, production planning, and equipment maintenance to improve manufacturing and maintenance sustainability. Caterino et al. [135] defined a new remanufacturing framework based on cloud computing technology called cloud remanufacturing (CRMfg). The framework translates remanufacturing resources and capabilities into services delivered via the Internet, allowing for the mutually beneficial connection of remanufacturing service providers and customers in different locations. The CRMfg can monitor registered remanufacturers and their equipment, thus enabling the scheduling and real-time tracking of product remanufacturing tasks. It can significantly improve the efficiency of remanufacturing, thus contributing to achieving economic and environmental sustainability. Çınar et al. [136] introduced the application of machine learning-based predictive maintenance in sustainable smart manufacturing. The collected real production data are used to train machine learning models for fault diagnosis, thus enabling predictive maintenance. Predictive maintenance can significantly reduce hidden problems, failures, and accidents in production, resulting in less breakdown maintenance and lower maintenance costs, ultimately achieving sustainable manufacturing.

From the above literature, it can be seen that the sustainability of the manufacturing process is improved after the introduction of modern technologies related to the Industrial Internet. Remote monitoring of equipment and production lines optimizes process parameters and reduces waste caused by suboptimal processes; optimizes production scheduling and reduces economic consumption caused by waiting time; and minimizes the frequency of failures and reduces damage caused by equipment downtime.

## 4. Industrial Application

Remote monitoring and maintenance technology has been widely implemented in the industrial area. This section demonstrates several real industrial application cases.

Yang et al. [23] demonstrate a case of monitoring and maintenance for a grinding and polishing robot. The robots are used for grinding and polishing prebaked carbon anodes, a key consumable for aluminum electrolysis production. The proposed framework enables data-driven maintenance, intelligent fault diagnosis and prediction, and knowledge-based maintenance and fault diagnosis service for the robots. The authors developed a B/S-architecture industrial internet service platform. The platform includes the following functions: maintenance of equipment according to the manufacturer's predefined maintenance schedule; identification of process parameter anomalies; monitoring of machining accuracy; and answering questions about equipment failures, process parameters, and machining accuracy through knowledge graph Q&A technology.

Li et al. [22] developed an Industrial 4.0 platform for equipment monitoring and maintenance in prebaked carbon anode production. This platform is applied for the monitoring and maintaining the kneader, critical equipment for prebaked carbon anode production. The platform can collect equipment working condition data, production process data, and product quality data, enabling production planning, equipment failure maintenance, process parameter optimization, and product quality control.

Scheuermann et al. [137] proposed an example of an Industrial 4.0 manufacturing process called the Agile Factory. The Agile Factory is implemented in mass customization production scenarios. The Agile Factory assembly line is component-based, combining trackable mobile workstations with fixed workstations. Therefore, the products are traceable during the production process. A customer feedback loop was implemented to allow for mass customization of products by permitting change requests during assembly time.

Bonci et al. [138] demonstrate a case study of the application of fault diagnosis technology in industrial packaging machinery. The collected current data of the equipment is pre-processed with a demodulating technique by means of the analytic envelope. The pre-processed signals are subjected to continuous wavelet transform and discrete wavelet transform for fault diagnosis. A real industrial case demonstrates that the method can detect belt failures in packaging machinery running continuously in non-stationary conditions.

As can be seen from the above cases, the main applications in the industry are equipment maintenance, process optimization, product quality control, production planning, troubleshooting, etc.

## 5. Challenges

Although a great effort has been dedicated to the monitoring and maintenance of equipment and production lines on the Industrial Internet, there are still many challenges that remain to be addressed. Key challenges stem from the requirements in real-time performance, interoperability, security, and intelligence. These challenges will be discussed in this section, as shown in Figure 11.

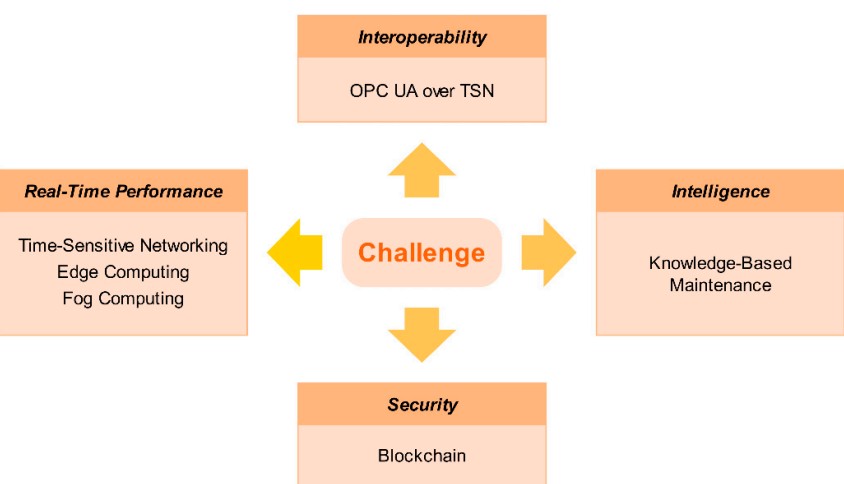

**Figure 11.** Challenges.

## 5.1. Real-Time Performance

In order to enable remote monitoring and maintenance of equipment and production lines in the Industrial Internet, sensors, actuators, machines, and other computing devices need to cooperate with each other. Low latency and reliable data transmission must therefore be ensured for the reliability and efficiency of the Industrial Internet. Real-time performance is a significant issue for data transport in the Industrial Internet [139].

The emergence of time-sensitive networks (TSN) holds the promise of solving this problem. Time-sensitive networking (TSN) is a set of standards under development by the time-sensitive networking task group of the IEEE 802.1 working group [140,141]. It achieves time synchronization, limited low latency, and high reliability over standard connection technologies such as Ethernet that meet the requirements of time-sensitive applications in industrial systems [142]. It has the potential to become the standard for the next generation of industrial communication and automation [11,143].

The National Institute of Standards and Technology (NIST) has built a collaborative robotic workcell testbed enabled by Wireless TSN technologies. The study shows that the percentage of idle time experienced by the operator robot is lower when TSN is enabled because the robot can receive commands from the controller more rapidly, which increases the productivity of this collaborative robot in industrial environments [144]. Yang et al. [145] proposed a TSN chain flow abstraction, TC-Flow, that solves the problem of coordinated scheduling of multiple data streams in industrial applications such as control and security applications. Nikhileswar et al. [146] present an industrial control system implemented by 5G and TSN and evaluate it. The results show that TSN can significantly reduce the latency of the network. Pop et al. [147] proposed that using TSN as a deterministic transport for the fog computing network layer in industrial automation can reduce the latency and improve the stability of data transmission in the Industrial Internet.

Overall, TSN combined with edge computing and fog computing is expected to be a way to improve the real-time performance of the Industrial Internet.

## 5.2. Interoperability

In the Industrial Internet, the interconnection of people, machines, and things is to be realized. The transmission of real-time data from industrial equipment and information from network applications such as operations management are separated in existing factory intranets, with the former generally being routed through Fieldbus or industrial Ethernet and the latter relying on conventional Ethernet. OPC UA has been able to interconnect devices at the application layer but not at the data link layer in the Open Systems Interconnection (OSI) model. TSN enables network interconnection and data interoperability at

the data link layer. Therefore, the convergence of OPC UA and TSN promises a unified Industrial Internet [8].

Pfrommer et al. [10] presented an approach that combines a non-real-time OPCUA server with a real-time OPC UA Pub Sub, where both have accessibility to the shared information model without dropping the real-time guarantee to the publisher. The publisher can therefore run within hardware-triggered interrupts to guarantee low latency and less jitter. Li et al. [12] present a two-layer manufacturing system communication architecture with OPC UA and TSN technology in heterogeneous networks. The TSN network is used as the communication backbone for realizing the real-time services of the industrial automation system, which connects the heterogeneous industrial automation subsystems at the field level with the upper-layer entities. OPC UA is used to realize the exchange of information between the heterogeneous subsystems in the field layer and the entities in the upper layers. The results prove that different types of devices can communicate with each other in this system with excellent real-time performance.

From the above, it can be seen that OPC UA over TSN will be a promising way to solve interoperability problems in the Industrial Internet.

### 5.3. Security

The main features of the Industrial Internet are openness, interconnection and sharing, which pose serious security challenges. An attack on the network can lead to loss, leakage, and tampering of industrial data. In the event of an attack on the control network, there would be enormous financial damage and even a threat to the lives of the public. It seems that blockchain holds the promise of alleviating these problems [148].

Gu et al. [149] implemented a functional safety and information security protection mechanism based on blockchain technology in the CPS system. The equipment must be authenticated Safety Integrity Level (STL) before accessing the CPS. A new equipment block is created with a combination of asymmetric and symmetric key encryption methods, and the STL of the equipment will be stored in the block. The researchers propose a fault threshold mechanism based on smart contract technology to ensure functional safety and information security during equipment communication. Qu et al. [150] combine blockchain and federated learning technologies to propose a blockchain-enabled federated learning (FL-Block) model which enables decentralized privacy protection. FL-Block enables decentralized privacy protection through hybrid identity generation, comprehensive authentication, access control, and off-chain data storage and retrieval.

It can be concluded that blockchain technology is emerging as a prospective solution to security issues in the Industrial Internet due to its decentralization, non-tamperability, traceability, and high cryptographic security.

### 5.4. Intelligence

Artificial intelligence is already widely used in the maintenance of equipment and production lines. The common approach is data-driven fault diagnosis, which can be used to diagnose and isolate faults in specific devices. However, the implementation of data-driven fault diagnosis requires careful design of physical models, signal patterns, and machine learning algorithms to describe faults [17]. A knowledge-based fault diagnosis approach is suitable for complex or multi-component systems/processes without detailed mathematical models, which is becoming another direction of development [151,152].

Cao et al. [153] proposed a novel Knowledge-based System for Predictive Maintenance in Industry 4.0 (KSPMI). KSPMI blends computational intelligence and symbolic intelligence. Firstly, chronicle mining (a special type of sequential pattern mining approach) is used to extract machine degradation models from industrial data. After that, domain ontologies and Semantic Web Rule Language (SWRL) rule-based reasoning use the extracted chronicle patterns to query and reason on system input data with rich domain and contextual knowledge. The system is able to predict future failures of equipment and the time of occurrence. Wang et al. [154] presented a framework for the intelligent operation

and maintenance of traction transformers based on knowledge graphs. The framework integrates multiple sources of heterogeneous data from traction transformers into structured knowledge with a unified knowledge representation model. The researchers constructed a knowledge entity graph, concept graph, fault treatment graph, and fault case graph to achieve multi-source condition data fusion and correlation analysis, multi-dimensional differentiated state evaluation, and intelligent assisted maintenance decision-making.

As can be seen, the combination of data-driven fault diagnosis and knowledge-driven fault diagnosis may lead to further progress in the intelligent maintenance of equipment in the future.

## 6. Conclusions

In this paper, a detailed overview of the monitoring and maintenance of equipment and production lines on the Industrial Internet is proposed. At first, a brief review of its architecture is presented, and a three-layer reference architecture is summarized, containing the physical layer, the transport layer and the application layer. We then provide a detailed literature review of the key enabling technologies involved in each layer, including data acquisition technologies, edge computing, communication technologies, fog computing, big data, artificial intelligence, digital twins, data analytics, O&M optimization, and sustainability. Next, we demonstrate some industrial application cases. We also discuss the challenges in terms of real-time performance, interoperability, security, and intelligence. Overall, we have reviewed the most advanced research in this field and discussed the direction of future research, which is expected to be a reference for researchers addressing this area.

There are still some limitations in this review because remote monitoring and maintenance for equipment and production lines on the Industrial Internet is a very wide field. In the Industrial Internet, many nodes work simultaneously and are prone to failures. Therefore, the fault tolerance of the system is a key issue in the face of various potential failures. Numerous sensors and devices continuously consume large amounts of energy, so research on an energy-efficient 'green Industrial Internet' is also necessary. There are also some traditional industrial software systems in industrial applications, such as enterprise resource planning (ERP), manufacturing execution systems (MES), etc. How to interoperate with these systems to further improve the digitalization and intelligence of production is a huge challenge. In the future, we will conduct special research on the above issues.

**Author Contributions:** Conceptualization, P.J. and Q.L.; Investigation, Q.L. and Y.Y.; Resources, Y.Y.; Writing—original draft preparation, Q.L.; Writing—review and editing, P.J.; Visualization, Q.L.; Supervision, P.J.; Project administration, P.J.; Funding acquisition, P.J. All authors have read and agreed to the published version of the manuscript.

**Funding:** This research was funded by the National Key Research and Development Program of China, grant number 2021YFE0116300; the National Natural Science Foundation of China, grant number 51975464.

**Institutional Review Board Statement:** Not applicable.

**Informed Consent Statement:** Not applicable.

**Data Availability Statement:** Not applicable.

**Conflicts of Interest:** The authors declare no conflict of interest.

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
