# Peer review of "Remote Monitoring and Maintenance for Equipment and Production Lines on Industrial Internet: A Literature Review"

_machines, doi:10.3390/machines11010012_

Round 1
Reviewer 1 Report
The paper deals with a literature review of the new technologies related to the Industrial Internet used in the maintenance field. It is interesting and very well written. My comments for improving the research are in the following:
1) In my opinion, considering the important climate change we are living, some aspects related to the sustainability should be considered in this paper. I suggest to introduce this aspect by citing some papers in the maintenance field that consider the sustainability. In such a way, the readers will have also clear the modern context.
I suggest in the following some papers you can add related to the topics of your research:
1) Rojek, I., Mikołajewski, D., & Dostatni, E. (2020). Digital twins in product lifecycle for sustainability in manufacturing and maintenance. Applied Sciences, 11(1), 31.
2) Caterino, M., Fera, M., Macchiaroli, R., & Pham, D. T. (2022). Cloud remanufacturing: Remanufacturing enhanced through cloud technologies. Journal of Manufacturing Systems, 64, 133-148.
3)Çınar, Z. M., Abdussalam Nuhu, A., Zeeshan, Q., Korhan, O., Asmael, M., & Safaei, B. (2020). Machine learning in predictive maintenance towards sustainable smart manufacturing in industry 4.0. Sustainability, 12(19), 8211.
2) Based on the previous comments, I also suggest authors to consider the impact of the introduction of these new technologies for improving maintenance from the sustainability point of view. In my opinion, the question authors should consider to answer is: "What is the impact of the introduction of the modern technologies related to the Industrial Internet in terms of sustainability (environmenta, economic and social impact)?"
I suggest authors to introduce few papers that consider this aspect in their paper.
3) In the conclusion section, I suggest to highlight the main limitations of the present paper and how authors think can be overcome in future developments of this research.
4) In the title you use the word "survey" to identify what you have done. In my opinion, this word can be misleading in this paper. In fact, usually, the word "survey" is used in researches that make questionnaire on a specific topic and submit it to companies. In my opinion, in this paper, authors carries out a literature review, not a survey. I suggest to revise the title accordingly.
Reviewer 2 Report
The authors proposed a survey on monitoring and maintenance for equipment and production lines on industrial internet. The paper has to be improved, in particular, the survey presents a lot of missing references, topics, and insights in the area of monitoring and maintenance for equipment and production lines on industrial internet. For this reason, the authors should address the following major points:
- The English should be improved since a lot of typos are in the manuscript. For example, Figure 6. Wireless Communication Technologies, "blueteeth should be "bluetooth" Line 451, "self-attentive mechanism" should be "self-attention mechanism" Please read the paper carefully.
- The descriptions of the technologies are not exhaustive, for example in the table "Comparison Between Wireless Communication Technologies", Low-Power Wi-Fi is not present as well Bluetooth LE, Z-Wave, NFC, RFID.
- I suggest dividing the column Power into 3 levels, High, Moderate, Low because in my opinion, Bluetooth is not High but Moderate if you compare it with Wi-Fi.
- In Big Data Application Framework, I suggest discussing about data ingestion and data streaming platforms such as Apache Kafka, MQRabbit and communication standards/protocols such as AMQP, MQTT etc.
- Technologies related to data storage are missing, for example, SQL DB, NoSQL DB, Data Warehouse, Data Lake, etc.
- In Application Section, I suggest adding applications related to Data Analytics and Optimizing Operations and Maintenance (O&M).
- Literature related to Fault Diagnosis and Anomaly Detection is poor. The inspiration of your work must be highlighted, for example in 10.1109/TASE.2021.3069109 the authors proposed a method for fault diagnosis with low computation effort that can be easily developed in industrial equipment, or in 10.1016/j.engappai.2015.05.004, the authors proposed a fault diagnosis methodology for quality control scenario in production lines.
- The paper should include more real applications related to "Monitoring and Maintenance" since, in this form, the paper seems a survey focused only on technologies.
Reviewer 3 Report
Thank you for the opportunity to review the paper entitled "Remote Monitoring and Maintenance for Equipment and Pro-2 duction Lines on Industrial Internet: A Survey".
First of all, the paper misses logical flow.
You need to rewrite the Introduction part to be organized to describe the needs, I'm, and research framework. Now, your Introduction looks more like a part of the Literature review.
In the literature review, you need to clarify the main concepts of your work, such as Architecture and Key Enable Technologies. To fill literature review please consult following literature
Digital Twin Testbed and Practical Applications in Production Logistics With Real-Time Location Data (http://doi.org/10.24867/IJIEM-2021-2-282)
Digital Servitization and Firm Performance: Technology Intensity Approach (https://doi.org/10.5755/j01.ee.33.4.29649)
Predictive Maintenance and Intelligent Sensors in Smart Factory: Review (https://doi.org/10.3390/s21041470)
You miss the part about Methodology, you need to explain your case, query, or data-set. Moreover, you overlook details about Results and Discussion. Without these parts, my suggestion is to reject the paper.
The conclusion part needs to summarise previous work, and give limitations and future implications.
Round 2
Reviewer 2 Report
Reading the paper again I noticed that another important aspect could be discussed a bit in the paper related to the IEC62541 standard, OPC-UA since it permits data exchange from sensors (field) to cloud applications.
Reviewer 3 Report
The all suggestions are incorporated, and paper could be published.
Author Response
Thank you very much for your recognition of our research.